# Antitumor Effect of Berberine Analogs in a Canine Mammary Tumor Cell Line and in Zebrafish Reporters via Wnt/β-Catenin and Hippo Pathways

**DOI:** 10.3390/biomedicines11123317

**Published:** 2023-12-15

**Authors:** Alessandro Sammarco, Giorgia Beffagna, Roberta Sacchetto, Andrea Vettori, Federico Bonsembiante, Giulia Scarin, Maria Elena Gelain, Laura Cavicchioli, Silvia Ferro, Cristina Geroni, Paolo Lombardi, Valentina Zappulli

**Affiliations:** 1Department of Comparative Biomedicine and Food Science, University of Padua, 35020 Legnaro, Italy; alessandro.sammarco6@gmail.com (A.S.); giorgia.beffagna@gmail.com (G.B.); roberta.sacchetto@unipd.it (R.S.); federico.bonsembiante@unipd.it (F.B.); giulia.scarin91@gmail.com (G.S.); mariaelena.gelain@unipd.it (M.E.G.); laura.cavicchioli@unipd.it (L.C.); silvia.ferro@unipd.it (S.F.); 2Department of Urology, Houston Methodist Research Institute, Houston Methodist Hospital, Houston, TX 77030, USA; 3Department of Cardiac, Thoracic, Vascular Sciences and Public Health, University of Padua, 35128 Padua, Italy; 4Department of Biotechnology, University of Verona, 37134 Verona, Italy; andrea.vettori@univr.it; 5Department of Animal Medicine, Production and Health, University of Padua, 35020 Legnaro, Italy; 6Naxospharma Srl, 20026 Novate Milanese, Italy; cristina.geroni@hotmail.it (C.G.); p.lombardi@naxospharma.eu (P.L.)

**Keywords:** berberine derivates, dog, mammary tumors, zebrafish, Hippo, Wnt, β-catenin

## Abstract

The heterogeneous nature of human breast cancer (HBC) can still lead to therapy inefficacy and high lethality, and new therapeutics as well as new spontaneous animal models are needed to benefit translational HBC research. Dogs are primarily investigated since they spontaneously develop tumors that share many features with human cancers. In recent years, different natural phytochemicals including berberine, a plant alkaloid, have been reported to have antiproliferative activity in vitro in human cancers and rodent animal models. In this study, we report the antiproliferative activity and mechanism of action of berberine, its active metabolite berberrubine, and eight analogs, on a canine mammary carcinoma cell line and in transgenic zebrafish models. We demonstrate both in vitro and in vivo the significant effects of specific analogs on cell viability via the induction of apoptosis, also identifying their role in inhibiting the Wnt/β-catenin pathway and activating the Hippo signals with a downstream reduction in *CTGF* expression. In particular, the berberine analogs NAX035 and NAX057 show the highest therapeutic efficacy, deserving further analyses to elucidate their mechanism of action more in detail, and in vivo studies on spontaneous neoplastic diseases are needed, aiming at improving veterinary treatments of cancer as well as translational cancer research.

## 1. Introduction

Human breast cancer (HBC) is the most common type of tumor in women, accounting for 31% of all diagnosed cancers [1]. Despite recent advancements in the detection, diagnosis, and treatment of HBC, the heterogeneous nature of this disease leads, in some cases, to therapy inefficacy [2]. For this reason, spontaneous animal models of HBC are highly important for studying the molecular mechanisms underlying the onset and progression of these tumors. They can serve as a useful intermediary between traditional preclinical models and human clinical trials and therefore are useful for the identification of new therapies [3,4].

Canine mammary tumors (CMTs) are a highly heterogeneous group of tumors representing the most common type of tumors in female dogs. Approximately 50% of diagnosed CMTs are malignant [5], but recently, an increase in malignant vs. benign tumors was observed with a trend similar to that also described in human oncologic patients [6]. CMTs are spontaneous in dogs as well as in women, and many anatomic, clinical, and histological features of CMTs are similar to those reported in HBC [7,8,9]. As companion animals, dogs share the environment and lifestyle of humans and become consequently exposed to many of the same carcinogens. Better than genetically modified mice, dogs have higher physiological similarities with humans, such as comparable telomerase activities [10] and a high incidence of spontaneous cancers [11], making them a valuable cancer model for HBC research [12]. This spontaneous animal model could be highly beneficial to translational HBC research, particularly for the identification of new therapeutic targets and the collection of highly predictive data that may accelerate human cancer clinical research.

In the last decade, the Wnt/β-catenin pathway, involving β-catenin and the adenomatous polyposis coli (APC) proteins, has emerged as an important player in many human tumor types, including HBC [13,14,15]. The reduced membrane expression of E-cadherin and β-catenin has been associated with a poor prognosis in feline and canine MTs as well as in HBC [16,17]. E-cadherin, in association with other adhesion molecules of the β-catenin class, binds to the actin cytoskeleton. This association is important for epithelial cell function and for tissue integrity [18]. Moreover, β-catenin has a double function as an adhesion molecule and as an activator of the Wnt/β-catenin pathway. Specifically, in normal cells, free cytoplasmic β-catenin is entrapped by a complex containing axin, glycogen synthase kinase 3 (GSK3), and APC that facilitates the phosphorylation of β-catenin, which consequently can be degraded. In tumor cells, this complex is not able to phosphorylate β-catenin, which, as a result, cannot be degraded [19,20]. To the best of our knowledge, only little information is known about the role played by the Wnt/β-catenin signaling pathway in CMTs [21,22,23].

Additionally, the Hippo pathway has been identified as highly important in HBC onset and progression. Cordenonsi and co-authors demonstrated that the activity of the transducer of the Hippo pathway TAZ is required to sustain tumor-initiation capacities and the self-renewal of cancer stem cells (CSCs) in the breast. TAZ protein level is elevated in CSCs and in poorly differentiated HBC with a poor prognosis [24,25]. To date, the role of the Hippo pathway in CMTs has not been deeply investigated or fully clarified [23,26,27,28].

Berberine (BBR) is a bitter-tasting isoquinoline quaternary alkaloid isolated from many kinds of plants such as *Hydrastis canadensis*, *Berberis vulgaris* and other plant species. A mainstay of Traditional Chinese Medicine, BBR is also in use in Ayurvedic and Native American herbal medicines for its microbial and anti-inflammatory properties [29,30]. BBR has multiple pharmacological properties, including antibacterial, anti-inflammatory, antidiabetic, hepatoprotective, and neuroprotective. It also controls the expansion of blood vessels and the inhibition of platelet aggregation and, therefore, has a wide spectrum of medical applications against, inter alia, gastroenteritis, hyperlipidemia, non-alcoholic fatty liver disease, coronary artery disease, hypertension, diabetes, and Alzheimer’s disease [31,32,33,34]. There is an increasing interest in the clinical efficacy of BBR, as manifested by almost 90 completed and ongoing clinical studies, including 5 for cancer indications. 

In vitro studies using HBC cell lines demonstrated that BBR inhibits cancer cell migration and proliferation and affects cell viability, inducing apoptosis [15,35,36,37]. More recently, it has been shown that BBR could be a promising drug able to suppress the cell growth and cell invasiveness of triple-negative HBC (TNBC) cells through IL-8-related mechanisms [38,39]. BBR was also demonstrated to be a promising drug in the prevention and treatment of colon cancer through the inhibition of Wnt/β-catenin signaling [40] and in exerting anticancer activity, delaying the development of MTs in transgenic mice for the HER-2/neu oncogene [41,42,43].

Berberrubine (BRR) is the first-pass metabolism product of BBR [44], and little is known about its antitumor properties [45,46]. BRR has been reported to induce topoisomerase II-mediated DNA cleavage [47] and to bind G-quadruplex nucleic acid structures [48].

The aim of our study was to evaluate the potential of BBR, BRR, and eight structurally related analogs (Figure 1) as anticancer agents. We conducted in vitro experiments to examine the biological effects of these compounds on a malignant CMT cell line in order to identify candidates that might become potential anticancer agents. The analogs under investigation possess aromatic groups attached to the 13-position of the parent alkaloid skeleton through a hydrocarbon linker [49]. Their effects on Wnt/β-catenin signaling [50] and on some relevant human tumor types have been reported, both in vitro [36,51,52,53,54,55] and in vivo [41,56].

To elucidate the mechanism involved in the antitumor effect of BBR, of BRR, and of selected analogs, we investigated the association with the Wnt/β-catenin signaling and Hippo pathways. More specifically, we used Wnt/β-catenin and Hippo signaling pathway reporter zebrafish lines to confirm the in vivo effect of these BBR and BRR analogs on these pathways. We expected to identify at least one analog able to kill cancer cells mainly by apoptosis and better clarify how the Wnt/β-catenin and Hippo signaling pathways are involved.

## 2. Materials and Methods

### 2.1. Chemicals

Berberine (BBR—chloride form, Trust & We, Shanghai, China), berberrubine (BRR), and analogs NAX012, NAX014, NAX035, NAX053, NAX057, NAX060, NAX085, and NAX118 were provided by Naxospharma (Figure 1). The purity (>95%) of the derivatives was previously assessed [51,52].

### 2.2. Cell Culture and Drug Treatment

The canine mammary carcinoma cell line CF33 (kindly provided by Dr. R. De Maria, University of Turin, Turin, Italy) was maintained in Dulbecco Modified Eagle Medium (DMEM, Thermo Fisher Scientific, Waltham, MA, USA) supplemented with 10% Fetal Bovine Serum (FBS, Thermo Fisher Scientific), 100 units/mL penicillin, and 100 μg/mL streptomycin (Thermo Fisher Scientific) under a humidified atmosphere of 5% CO_2_ at 37 °C. BBR, BRR, and analogs were dissolved in dimethyl sulfoxide (DMSO, Thermo Fisher Scientific), aliquoted, and stored at −20 °C with a stock concentration of 40 mM. The final concentration of DMSO did not exceed 0.1%.

For the in vitro experiments, CF33 cells were plated and treated separately with BBR, BRR, NAX012, NAX014, NAX035, NAX053, NAX057, NAX060, NAX085 and NAX118. Two different experiments were performed. In the first experiment, we used different drug dosages, from 2 μM to 40 μM, for a steady incubation time of 24 h. In the second experiment, we used the same drug dosage, 10 μM, for different incubation times (from 3 to 24 h). Control cells were incubated with a respective maximal percentage of DMSO alone (<0.1%). The two experiments were performed in triplicate.

### 2.3. Cell Viability Assay

CF33 cells were plated in a 96-well plate at a density of 1 × 10^4^ cells/well in 100 μL of complete medium. After 24 h, a medium containing the specific drug was added. To measure cell viability, cells were incubated with the different drugs, and the in vitro toxicology assay kit TOX8 (Sigma-Aldrich, St. Louis, MO, USA) was used following manufacturer’s instructions. The experiment was performed in triplicate. Experimental results were expressed as percentage of cell viability in comparison with control DMSO-treated cells. 

### 2.4. Quantification of Necrosis/Apoptosis by Flow Cytometry

In order to assess the rate of necrosis and apoptosis induced by BBR, BRR, and analogs at the same dosage (10 µM) and at the different incubation times (6, 12, 18, and 24 h of incubation), two biological replicates (both performed in technical triplicate) of CF33 were analyzed by flow cytometry using the Annexin V-Fluorescein isothiocyanate (FITC) Apoptosis Detection Kit (eBioscience, Thermo Fisher Scientific, Waltham, MA, USA) following the manufacturer’s instructions. Briefly, cells were trypsinized and centrifuged at 1100 rpm for 10 min at 4 °C. Cells were resuspended in 200 µL of binding buffer with 5 µL of annexin V–FITC and incubated for 10 min in the dark at room temperature. After the incubation, 200 µL of binding buffer was added to the cells that were subsequently centrifuged at 1100 rpm for 10 min at 4 °C, and the supernatant was discarded. Then, 10 µL of Propidium Iodide (PI) and 900 µL of binding buffer were added. Samples were incubated for 5 min in the dark at room temperature. Cells were then acquired using the flow cytometer CyFlow Space (Partec-System, Sysmex Europe GmbH, Norderstedt-Amburgo, Germany). Data were analyzed using FlowJo (version 10, BD, Franklin Lakes, NJ, USA) or the open-source software FCSalyzer (version 0.9.16-alpha). Late apoptotic/necrotic cells were positive for both Annexin-V FITC and PI, while apoptotic cells were positive for Annexin-V FITC only.

### 2.5. Protein Extraction and Western Blot Analysis on CF33 Cells

At the end of incubation time (24 h), CF33 cells treated with BBR, BRR or analogs at a final dosage of 2 μM were washed twice with ice-cold phosphate-buffered saline (PBS), and total lysate was obtained by solubilization in 5% sodium deoxycholate supplemented with protease inhibitors (Sigma-Aldrich). Protein concentration was determined by the bicinchoninic acid–protein assay (Pierce, Appleton, WI, USA), using bovine serum albumin as standard. Protein fractions were stored at −80 °C. Proteins were resolved by SDS-PAGE using 7.5% polyacrylamide gels [57], and the same amount of protein (20 μg) was loaded for all the samples. After electrophoresis, proteins were transferred onto nitrocellulose, which was stained with Ponceau S (Sigma-Aldrich). The proteins were transferred to the membrane and blocked in 5% skim milk before being probed overnight at 4 °C with the primary antibodies. The blots were probed with the following antibodies: rabbit polyclonal antibodies specific for WW domain-containing transcription regulator protein 1 (WWTR1) (1:1000, Sigma-Aldrich #HPA007415), recognizing both YAP and TAZ human proteins; rabbit monoclonal antibodies to non-phosphorylated (active) β-Catenin (1:1000, Cell Signaling, Danvers, MA, USA #8814); and mouse monoclonal antibodies to β-catenin (1:1000, BD Biosciences, New York, NY, USA #610154). After washing the membrane, an HRP-conjugate secondary antibody was added and incubated with the membrane for 1 h at room temperature. The membrane was washed and analyzed by a chemiluminescence analyzer using the chemiluminescence ECL reagent.

### 2.6. RT-PCR and Semi-Quantitative PCR

Total RNA was isolated from control CF33 cells and CF33 cells 24 h after treatment with 2 μM of drug using an RNeasy Micro Kit (Qiagen, Hilden, Germany) following the supplier’s protocol. The obtained RNA was then quantified using Qubit Fluorometric Quantitation (LifeTechnologies, Carlsbad, CA, USA), and 500 ng was reverse transcribed using RevertAid First Strand cDNA Synthesis Kit (Thermo Fisher Scientific), according to the manufacturer’s instructions. Then, 5 μL of cDNA products was amplified with 1 unit of GoTaq (Promega, Madison, WI, USA) in the buffer provided by the manufacturer containing MgCl2 with dNTPs and in the presence of the specific primers for *β-catenin*, *YAP*, *TAZ*, *CTGF*. *Actin-beta* was used as a housekeeping gene. Primers and PCR conditions are summarized in Table 1. A first cycle of 2 min at 95 °C was followed by 30 s at 95 °C, 30 s at 58 °C and 30 s at 72 °C for 27 cycles. The used number of cycles was chosen so that none of the samples reached a plateau at the end of the amplification protocol; therefore, all the samples were in the exponential phase of amplification. Each set of reactions always included a negative control, where the sample was replaced by nuclease-free water and was performed in triplicate.

### 2.7. Gel Electrophoresis, Acquisition of Gel Images and Quantitative Analysis

The PCR products were loaded onto a 2% agarose gel stained with gel-red (Promega, Milano, Italy). A 100 bp DNA ladder molecular weight marker (Life Technologies, Rockville, MD, USA) was run on each gel to confirm the expected molecular weight of the amplification product. Images of the RT-PCR agarose gels were acquired with BioRad gel imaging systems and band quantification was performed using ImageJ software (version 1.53s). The ratio between the gene of interest and actin-beta was calculated to normalize for initial variations in sample concentration and as a control for reaction efficiency. The mean and standard deviation of all experiments performed were calculated after normalization to actin-beta.

### 2.8. Ethics Statements

Zebrafish embryos and adults were raised, staged, and maintained at the Zebrafish Facility of the University of Padua, under standard conditions [58]. All husbandry and experimental procedures complied with European Legislation for the Protection of Animals used for Scientific Purposes (Directive 2010/63/EU) and with Italian law on animal experimentation (D.L. 4 March 2014, n.26). All the procedures were carried out under authorization n. 407/2015-PR from the Italian Ministry of Health. The project was also examined and approved by the Ethics Committee of the University of Padua with protocol number 18746.

### 2.9. Zebrafish Housing and Maintenance

Zebrafish used for all the experiments were taken from the breeding stocks of the Zebrafish Facility of the University of Padua and fed four times a day on a variable diet of dried food and live Artemia. Fish were kept in a 14:10 light:dark cycle in a 10 L multi-tank constant flow system. Water temperature was 28.5 °C and water was replaced at a rate of 10% per day. Eggs were collected and transferred in 10 cm petri dishes. The following day, dead eggs were removed, and dishes were cleaned. Zebrafish embryos were kept at 28.5 °C in system water mixed with methylene blue (2 mL 0.1% methylene blue per 1 L). Humane endpoints were not used considering that all the experimental procedures were performed in wild-type and transgenic larvae 5 days post fertilization. For the anesthesia or euthanasia of zebrafish embryos and larvae, Tricaine was added to the fish water at 0.16 or 0.3 mg/mL, respectively. The study was carried out in compliance with the ARRIVE guidelines as reported by Percie du Sert and colleagues [59]. All the experimental procedures were performed by specifically trained personnel—the veterinary staff working in the Zebrafish Facility of the University of Padua.

### 2.10. Zebrafish Tg(7xTCF-Xla.-Siam:mCherry) and Tg(Hsa.CTGF:mCherry) Transgenic Lines

In order to confirm the in vivo effect of BBR and NAX035, NAX053 and NAX057 on the Wnt/β-catenin and Hippo pathways, we used two zebrafish transgenic lines named Tg(7xTCF-Xla.-Siam:mCherry) [60] and Tg(Hsa.CTGF:mCherry), respectively [61]. Fish of all strains, maintained in Padua Zebrafish Facility, were monitored daily for the presence of signs of sickness, pain, distress, suffering, or moribund conditions; all treated zebrafish larvae were euthanized before the phenotypic analysis.

### 2.11. LD50

The median lethal dose (LD50) of BBR and analogs identified as the drug dosage able to kill 50% of 24 h post-fertilization (hpf) zebrafish larvae, incubated with the drugs for 24 h, was calculated with the Quest Graph™ LD50 Calculator, (AAT Bioquest, Inc., Sunnyvale, CA, USA). Different dosages of drugs were directly added to the fish water in 96-well plates. For each treatment, we used at least 96 embryos, and the experiment was performed in triplicate.

### 2.12. In Vivo Drug Treatments

Zebrafish embryos were incubated with 100 μM BBR (BBR LD50) and 10 μM of NAX035, NAX053 and NAX057 (LD50) at 24 hpf for 24 or 48 h. Drugs were added directly to the fish water in six-well plates. For each treatment performed in triplicate, at least 10 embryos were used.

### 2.13. Microscopy and Image Acquisition

The mCherry-expressing embryos belonging to the transgenic lines named Tg(7xTCF-Xla.-Siam:mCherry) and Tg(Hsa.CTGF:mCherry) were analyzed using a Leica M165FC epifluorescent microscope. All pictures were acquired with a Leica DC 500 digital camera, and contrast and brightness were elaborated with Adobe Photoshop 6.0 software. In each transgenic embryo, to quantify the level of fluorescence (that is strictly associated with the activity of Wnt/β-catenin of Hippo pathways), the integrity density was calculated using ImageJ software (version 1.53s). With ImageJ, we were able to analyze the embryos’ fluorescence in the head, cardiac region, and spinal cord. The yolk sac was not included since it usually shows autofluorescence after drug treatment. 

### 2.14. Statistical Analysis

Statistical analysis was performed using GraphPad Prism 9 software. Mean differences among groups were analyzed using the one-way ANOVA with Tukey’s multiple comparison test when data were normally distributed. When data were not normally distributed, the Kruskal–Wallis test with Dunn’s multiple comparison test was used. The Shapiro–Wilk test was used to test for normal distribution. The level of significance was set at *p* < 0.05.

## 3. Results

### 3.1. BBR, BRR, and Analogs Induce a Dose-Dependent Inhibition of Tumor Cell Viability

To evaluate the in vitro antiproliferative effect of BBR, BRR, and analogs (Figure 1) on CF33 cells, we measured cell viability after treatment. In the first experiment, cells were treated with BBR and analogs for 24 h using different dosages (2 μM, 5 μM, 10 μM, 20 μM, and 40 μM) (Figure 2a). The half maximal inhibitory concentration (IC50) was analyzed for each compound and is reported in Appendix A. Consequently, we performed a second cell viability assay at different time points, using a dosage of 10 μM for all the compounds (Figure 2b). In this experiment, cells were treated with each drug at the dosage of 10 μM, and cell viability was measured at different times (3, 6, 12, 15, 18, and 24 h) (Figure 2b). Treatments with BBR, BRR, and analogs exerted a dose- and time-dependent inhibition of cell viability (Figure 2).

NAX012, NAX035, NAX053, NAX057, and NAX060 were identified as promising drug candidates able to kill more than 50% of tumor cells after 3 h. NAX035, NAX053, NAX057, and NAX060 were able to kill more than 85% of tumor cells after 24 h (Figure 2b) and were selected for further analyses. BRR was also included as a comparison, being an apparently less efficient compound.

### 3.2. Induction of Necrosis/Apoptosis in Canine Mammary Tumor Cells by BBR Analogs

Next, we investigated whether the cytotoxic effects of BBR and analogs were due to apoptotic or necrotic cell death. CF33 cells were exposed to 10 μM of each drug, and early apoptosis and late apoptosis/necrosis were measured after 6, 12, 18, and 24 h of treatment by flow cytometry (Figure 3). From 6 to 12 h, an increased rate of early apoptosis versus necrosis was observed for most drugs (e.g., BBR, NAX012, NAX014, NAX035, NAX053, NAX057, NAX060, BRR, NAX085) (Figure 3). From 12 to 18 h, while some drugs (e.g., BBR, NAX012, NAX014, BRR) showed an even higher rate of early apoptosis versus necrosis, other drugs (e.g., NAX035, NAX053, NAX057, NAX060, NAX085) showed an increased rate of late apoptosis/necrosis. After 6 h, NAX053 and NAX057 induced a significantly higher amount of late apoptosis/necrosis when compared to the control (DMSO-treated cells). Similarly, after 12 h, treatment with NAX053 resulted in a significantly higher amount of late apoptosis/necrosis when compared with the control. After 18 h, BBR, NAX012, NAX014, and BRR induced a significantly higher amount of early apoptosis when compared with DMSO-treated cells. Finally, after 24 h, NAX085 caused a significantly higher amount of early apoptosis than control cells (Figure 3). Altogether, these data show a different time-dependent induction of apoptosis/necrosis among the different compounds.

### 3.3. NAX035, NAX053, NAX057, and NAX060 Induce a Down-regulation of Wnt/β-Catenin and an Activation of the Hippo Signaling Pathways in CF33 Cells

To evaluate the effect of BBR and the most efficacious analogs on the Wnt/β-catenin and Hippo pathways, we analyzed the protein expression of total β-catenin, active β-catenin, YAP, and TAZ after treatment. 

Western blot analysis revealed a decreased amount of total β-catenin in CF33 cells treated with NAX035, NAX053, NAX057, and NAX060. Interestingly, cells treated with the same compounds showed a notably lower expression of active β-catenin when compared to the controls and other analogs, suggesting a down-regulation of the Wnt/β-catenin pathway (Figure 4).

Considering the Hippo pathway, Western blot analysis also revealed a down-regulation of YAP and TAZ proteins in cells treated with NAX035, NAX053, NAX057, and NAX060 when compared to the control, which was more evident for TAZ (Figure 4). These results suggest a down-regulation of the Wnt/β-catenin and an activation of the Hippo pathway, respectively, when cells are treated with NAX035, NAX053, NAX057, and NAX060, when compared to the control cells. 

We focused on genes specifically involved in the Wnt/β-catenin and Hippo pathways. *β-catenin* mRNA expression was not significantly altered after treatment with NAX035, NAX053, NAX057 and BRR in comparison with cells treated with BBR and control. *β-catenin* mRNA expression was significantly higher (*p* < 0.01) in cells treated with NAX060 when compared to the control (Figure 5a).

Interestingly, *YAP* mRNA expression was significantly higher in cells treated with BBR, NAX053, and NAX060 when compared to the control (*p* < 0.05) (Figure 5b), whereas *TAZ* mRNA expression was not altered after treatment (Figure 5c). We also looked at the mRNA expression of *CTGF*, which is regulated by the Hippo pathway. Notably, *CTGF* mRNA expression was significantly lower in cells treated with NAX035 (*p* < 0.01) and NAX057 (*p* < 0.001) when compared to the control (Figure 5d), indicating an activation of the tumor-suppressor Hippo pathway after treatment with these BBR analogs.

### 3.4. BBR and NAX057 Reduce the Activity of Wnt/β-Catenin and Activate Hippo Signaling Pathways in Zebrafish Embryos

Next, we used zebrafish embryos to study the toxicity response to BBR, BRR and analogs by assessing the Median Lethal Dose (LD50) at 24 and 48 h of drug treatment. The LD50 of BBR, BRR, NAX085, and NAX118 was defined at 200 μM. NAX012 and NAX014 had an LD50 of 100 μM. NAX035 and NAX060 had an LD50 of 20 μM. Finally, NAX053 and NAX057 had an LD50 of 10 μM (Appendix A). For all tested compounds, zebrafish embryo LD50 values were not dependent on the duration of exposure, such that longer exposures (48 h) were not associated with lower LD50 values.

We tested BBR, NAX035, NAX053, and NAX057 on zebrafish Tg(7xTCF.Xlasiamois:nlsmCherry) and Tg(Hsa.CTGF:mCherry) transgenic lines, in which the expression of the reporter gene mCherry is regulated by specific cell signaling pathway-responsive elements [52,53]. By fluorescent microscopy imaging, we analyzed the amount of fluorescence indicating the expression of specific target genes in treated fish compared with controls. The analysis of these responsive transgenic lines, in which the expression of the reporter protein mCherry is directly associated with the activity of the canonical Wnt/β-catenin and Hippo signaling pathways, is simple and relatively immediate. In these in vivo experiments, we decided to use the drug dosages of 100 μM for BBR, 10 μM for NAX035, and 5 μM for NAX053 and NAX057 because these dosages were tested on zebrafish and did not induce any alteration in fish morphology and development. Indeed, fluorescence analysis and quantification need to be performed on animals that are correctly formed and developed. Tg(7xTCF.Xlasiamois:nlsmCherry) reporter animals treated with BBR, NAX035, NAX053, and NAX057 showed a significantly lower fluorescence when compared to the control (*p* < 0.0001) (Figure 6a), suggesting that these compounds down-regulated the canonical Wnt/β-catenin pathway in vivo.

Interestingly, Hippo pathway reporter animals showed a lower fluorescence when treated with BBR (*p* < 0.001), NAX035, and NAX057 (*p* < 0.05) when compared to the control (Figure 6b), while transgenic embryos treated with NAX053 showed an increased fluorescence compared to untreated siblings (*p* < 0.0001) (Figure 6b). 

## 4. Discussion

HBC is the most common invasive cancer in women. The treatment of HBC depends on the subtype of breast cancer, the stage of the disease, sensitivity to hormones, the patient’s age, and overall health [1]. The main treatment options include radiation therapy, chemotherapy, surgery, and hormone therapy. With treatment, a woman who receives a diagnosis of HBC has a percentage chance of surviving for at least 5 years ranging from almost 100 to 22 percent, depending on the stage of the disease at diagnosis and the tumor subtype [62]. For this reason, new treatment strategies are needed.

In recent years, phytochemicals, considered as bioactive ingredients present in plant products, became interesting drugs to study for their antitumor activity. This effect was demonstrated both in vitro and in vivo [63]. BBR is a phytochemical that has been reported to inhibit carcinogenesis in rats and mice [64] and in certain types of human cancers, as revealed by both in vitro and in vivo studies [38,41,65,66] and one clinical administration to human patients [40].

Regarding breast cancer, most in vitro studies on HBC cell lines have demonstrated BBR antitumoral effects with several involved molecules and pathways, mainly regarding cell proliferation and cell cycle, apoptosis, autophagy, and metastasis [65,67]. Only one study investigated BBR effects in vitro on a canine mammary tumor cell line (CF41.Mg), showing a decrease in cell viability after 24 h of BBR treatment (100 μM) [67]. In vivo mouse models also showed a reduction in tumor volume, tumor weight, or vessel density after BBR exposure [41,66,68,69,70,71,72,73]. In a recent study, Pierpaoli and collaborators explored the efficacy of the oral administration of a BBR derivative (NAX014) in a mouse model of HER2-overexpressing breast cancer. The authors showed no signs of toxicity after the oral administration of a high dose of the compound, suggesting its safety [56].

A major challenge today is to design novel drugs that target tumor cells specifically, with minimal cytotoxic effects on normal cells. Marverti and collaborators showed a prominent decrease in cell viability on two ovarian cancer cell lines, but a minimal effect on normal cells [74], when treated with BBR. This might indicate that BBR specifically targets tumor cells.

In this study, we chose eight semi-synthetic BBR analogs to investigate their effects on the CMT CF33 cell line for the first time, in comparison with BBR. In order to study the in vivo effect of these compounds, we used a transgenic zebrafish model. Despite this preliminary study being performed with a simple methodology, this type of comparison of the two models has never been performed before for CMTs. Our data demonstrated that the BBR analogs NAX035, NAX053, NAX057, and NAX060 affected cell viability, exerting a higher cytotoxic effect on cancer cells acting at a lower dosage in comparison with BBR. Generally, most of the tested drugs increased cell death by inducing cell apoptosis, which is a desirable effect for novel anticancer drugs. Apoptosis is a programmed cell death often overcome by tumors, whereas necrosis is often present in tumors and can be associated with increased metastases [75]. In line with our results, the BBR capacity to induce apoptosis is well known [65]. Wang and co-authors demonstrated that BBR inhibits proliferation and induces apoptosis of human cervical cancer (HeLa229) cells in a dose- and time-dependent manner, by the up-regulation of *p53* and the down-regulation of *Bcl2* and *Ptgs2* mRNA expression levels [76]. Additionally, the pro-apoptotic activity of BBR is well-documented in various breast cancer cell lines and many other cell cancer types [35].

In recent years, several data have emerged regarding the involvement of Wnt/β-catenin signaling in tumors. In particular, the increase in Wnt/β-catenin signaling seems to be important for human cancer onset and progression, including tumor initiation, tumor growth, cell death, cell differentiation and metastasis onset [77]. In this study, we showed that BBR and other BBR analogs down-regulated Wnt/β-catenin signaling. Our data were confirmed both at the protein level in vitro, by a decreased expression of active β-catenin in the tested canine cell line, and in vivo in zebrafish. In other studies, BBR has been found to deregulate Wnt/β-catenin signaling, inhibiting the proliferation, migration, and invasion of HBC cells in vitro [78], and human colorectal cancer growth both in vitro and in vivo in a mouse model [79,80]. Similarly, it was found by clinical administration that BBR potently attenuated intestinal polyps in familial adenomatous polyposis human patients via the inhibition of Wnt signaling [40]. In our study, we showed that at the gene expression level β-catenin mRNA, instead, did not show a significant decrease after drug treatment, except for cells treated with NAX060, which curiously presented a significant increase in comparison with control cells. We can speculate that the down-regulation of active β-catenin could be due to post-transcriptional or post-translational mechanisms. To date, it is well known that the mechanism whereby nuclear β-catenin drives or inhibits the expression of Wnt target genes is more diverse and less characterized [81]. Additional studies will be necessary to better understand the mechanistic underpinnings of the active β-catenin-observed plasticity. A deeper and more comprehensive characterization of the protein networks that regulate β-catenin transcription using zebrafish as a model to dissect molecular mechanisms involved in tumorigenesis could possibly drive us to confirm the identification of attractive new therapeutic targets.

The Hippo pathway is a well-conserved signaling pathway able to regulate organ size and tissue homeostasis [82]. This pathway can be down-regulated, leading to oncogenesis, through a variety of mechanisms [83]. These mechanisms include the induction of hyperproliferation, cellular invasion, and metastasis, and might play a role in cancer cell maintenance and chemotherapy resistance [84]. Our in vitro study on zebrafish for the Hippo pathway indicated an effect for BBR and other analogs on this signaling pathway. Particularly, there was a decrease in YAP and, mainly, TAZ proteins in cells treated with NAX035, NAX053, NAX057, and NAX060. Notably, the mRNA expression of the downstream gene *CTGF* was lower in cells treated with NAX035 and NAX057, when compared to the control. The Hippo pathway has already been found to be implicated in canine mammary carcinogenesis, and the described post-transcriptional/post-translational modifications of YAP and TAZ proteins justify the changes at the protein level, but not the changes at the mRNA level that were found in our and other studies [23,85]. Mechanistically, YAP and TAZ accumulate within the cytoplasm to translocate into the nucleus and activate downstream tumor-promoting genes when the Hippo pathway is switched off, whereas decreased cytoplasmic YAP/TAZ levels, due to their phosphorylation and degradation, are seen when the tumor-suppressing Hippo pathway is activated [86]. More recently, a Hippo-independent regulation of YAP/TAZ has been described, which could possibly explain non-decreased YAP levels in our study [87]. Several anticancer mechanisms of BBR-related compounds have been extensively studied [88], but only one study detected cell cycle arrest and apoptosis induced by a novel synthetic cyclizing-berberine on human cancer cell lines, which activated YAP phosphorylation [89]. No data are published on this Hippo-related anticancer effect of BBR analogs.

The analysis of the reporter gene expression in zebrafish treated for 48 h, particularly with BBR and NAX057 (NAX035 with no statistical significance) confirmed a down-regulation of CTGF. These data suggest an activation of the Hippo pathway, in comparison with controls, as also supported by a significant decrease in *CTGF* mRNA expression when cells were treated with NAX035 and NAX057. Instead, treatment with NAX053 indicated an in vivo up-regulation of CTGF. On one side, human studies have shown that CTGF can have pro-tumorigenic or anti-tumorigenic effects in different situations [90,91]. On the other hand, CTGF regulation is not controlled exclusively by the Hippo pathway [92]. Additional analysis to better characterize the response to these drugs in the in vivo zebrafish model should be performed, measuring the expression of other genes and proteins.

Further studies will be necessary to clarify the interesting interaction between Wnt/β-catenin and Hippo pathways in response to BBR and other analogs. We have only touched the tip of the iceberg in terms of understanding the intricacies and interconnections of the Wnt/β-catenin and Hippo pathways, which are responsible for the onset and progression of cancer proliferation, particularly in canine mammary tumors.

## 5. Conclusions

In conclusion, our results demonstrated that NAX035 and NAX057 could be considered promising anti-tumoral drugs. In short, (i) they kill approximately 50% of tumor cells at a low dose, (ii) they induce cell death mainly by apoptosis, (iii) they down-regulate the Wnt/β-catenin pathway, and (iv) interestingly, they also seem to act on the activation of the well-known tumor-suppressor Hippo pathway. Our results, obtained on a canine mammary tumor cell line, are in line with similar results demonstrated only for BBR in in vitro and in vivo human cancer studies, but with less toxicity also proved in our in vivo zebrafish model. Considering many similarities between canine mammary tumors and human breast cancer, these BBR analogs might represent relevant candidates to be tested in further in vivo animal models.

## Figures and Tables

**Figure 1 biomedicines-11-03317-f001:**
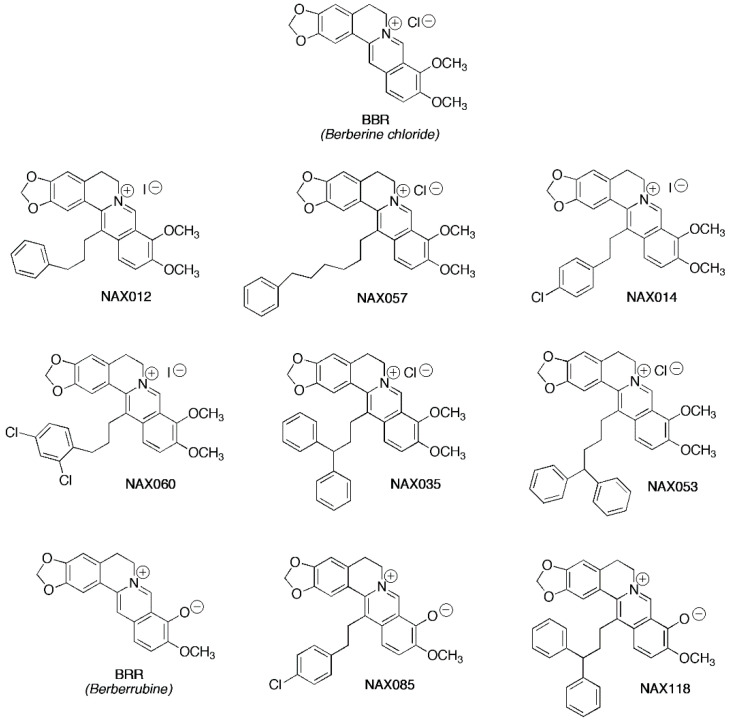
Biochemical structure of berberine (BBR), berberrubine (BRR), and analogs.

**Figure 2 biomedicines-11-03317-f002:**
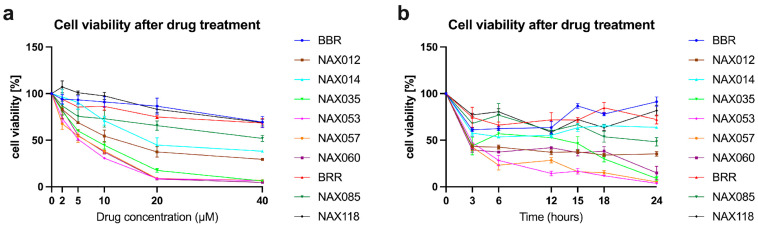
**Effect of berberine (BBR), berberrubine (BRR) and other analogs on CF33 cell viability.** (**a**) Cells were treated with 2 μM, 5 μM, 10 μM, 20 μM, and 40 μM of the different compounds for 24 h. (**b**) Cells were treated with 10 μM of the different compounds and cell viability was measured at different time points (3, 6, 12, 15, 18, and 24 h). BBR, BRR and analogs exerted a dose- and time-dependent inhibition of cell viability.

**Figure 3 biomedicines-11-03317-f003:**
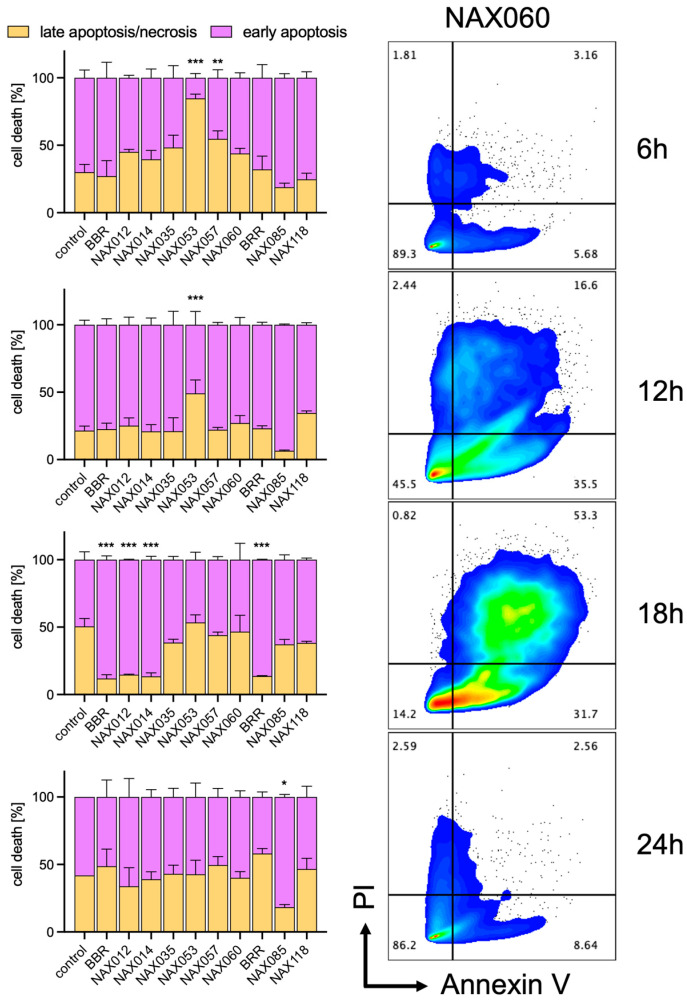
**Cell death after drug treatment.** Early apoptosis and late apoptosis/necrosis rates of CF33 cells treated with 10 μM of the different compounds at 6 h, 12 h, 18 h, and 24 h, measured using flow cytometry with double staining for annexin V and propidium iodide (PI). Early apoptosis cells were Annexin V^+^/PI^−^, whereas late apoptosis/necrosis were Annexin V^+^/PI^+^. *, *p* < 0.05; **, *p* < 0.01; ***, *p* < 0.001.

**Figure 4 biomedicines-11-03317-f004:**
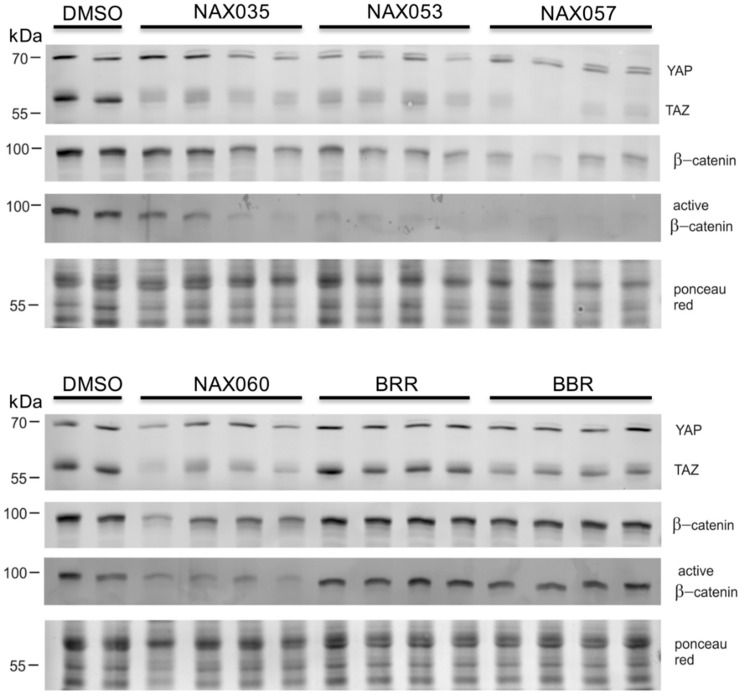
**Protein expression after drug treatment.** Representative cropped Western blots of YAP/TAZ, β-catenin and active β-catenin in protein lysates obtained from CF33 cells treated with berberine (BBR), NAX035, NAX053, NAX057, NAX060 and berberrubine (BRR) for 24 h. Bands of the expected sizes for YAP (~70 kDa), TAZ (~55 kDa), β-catenin (~92 kDa) and active β-catenin (~92 kDa) are present in all samples. The experiment was performed in quadruplicate. Each lane corresponds to a replicate. Ponceau red was used as a loading control.

**Figure 5 biomedicines-11-03317-f005:**
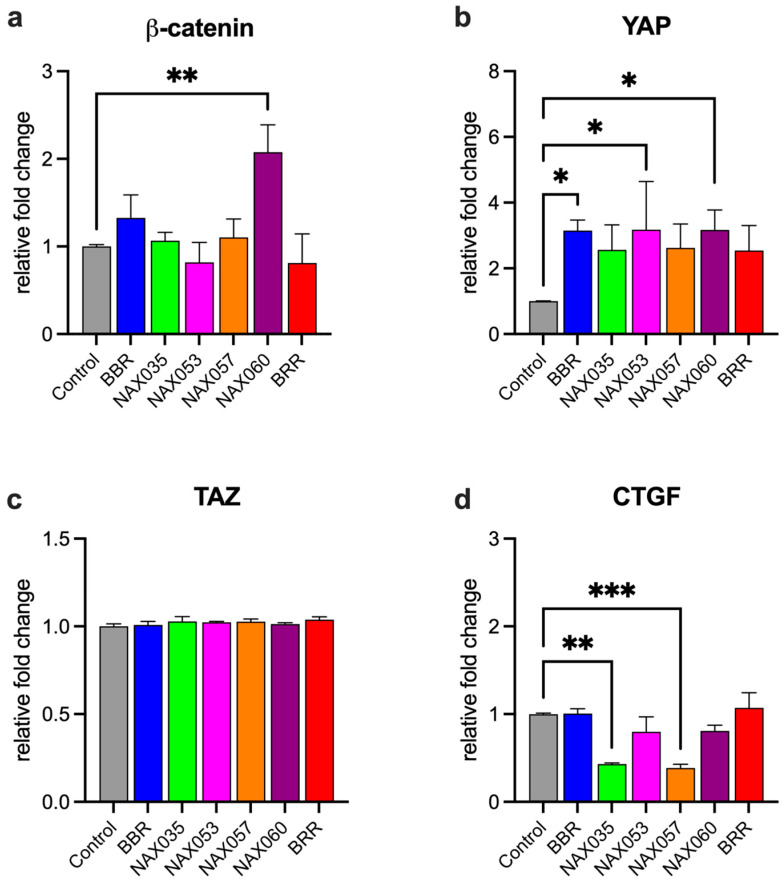
**mRNA expression after drug treatment.** Gene expression analysis by semi-quantitative PCR of genes directly or indirectly involved in Wnt/β-catenin and Hippo pathways in CF33 cells treated with berberine (BBR), NAX035, NAX053, NAX057, NAX060 and berberrubine (BRR). (**a**) *β-catenin*, (**b**) *YAP*, (**c**) *TAZ*, and (**d**) *CTGF* mRNA expression levels in CF33 cells. Relative gene expression levels are shown following normalization with beta-actin. *, *p* < 0.05; **, *p* < 0.01; ***, *p* < 0.001.

**Figure 6 biomedicines-11-03317-f006:**
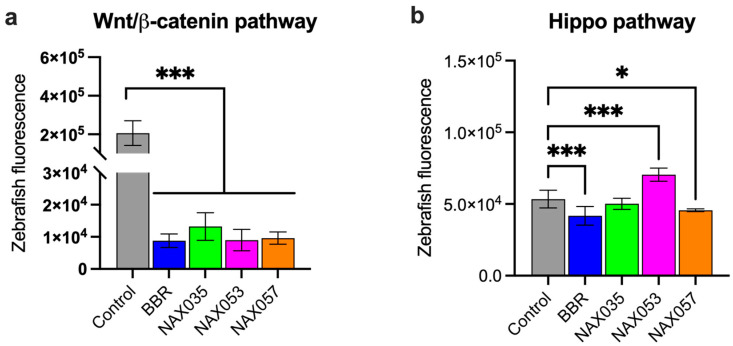
**Wnt/β-catenin and Hippo pathways zebrafish reporters**. Tissue specificity expression of mCherry under control of (**a**) *TCF/LEF* and of (**b**) *CTGF* genes. Quantification of mCherry fluorescence in control transgenic zebrafish in comparison with transgenic zebrafish treated with berberine (BBR) and other analogs (NAX035, NAX053, NAX057). *, *p* < 0.05; ***, *p* < 0.001.

**Table 1 biomedicines-11-03317-t001:** Primers and condition used for the semi-quantitative PCR analysis.

	Primer F (5′-3′)	Primer R (5′-3′)	Amplicon Length (pb)	Number of Cycles
ACTB	TGGCACCACACCTTCTACAA	CCAGAGGCGTACAGGGATAG	182	25
β-catenin	ACACGTGCAATCCCTGAACT	CACCATCTGAGGAGAACGCA	138	26
TAZ	TCCAATCACCAGTCCTGCAT	AGCTCCTTGGTGAAGCAGAT	125	28
YAP	CCCAGACTACCTTGAAGCCA	CTTCCTGCAGACTTGGCATC	107	28
CTGF	CGACTGGAAGACACGTTTGG	AGGAGGCGTTGTCATTGGTA	136	27

## Data Availability

Data are contained within the article and supplementary materials.

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
