# Peer review of "Antitumor Effect of Berberine Analogs in a Canine Mammary Tumor Cell Line and in Zebrafish Reporters via Wnt/β-Catenin and Hippo Pathways"

_biomedicines, 2023, doi:10.3390/biomedicines11123317_

Round 1

Reviewer 1 Report

Comments and Suggestions for Authors

The submitted work describes the antitumor effect of berberine and some of its analogues, total 10 compounds. The article is well-written, the amount of new data is significant, the discussion is complete. I also really enjoyed the introduction – clear, concise.

However, I would also like to request some changes, listed below.

First, in the title “novel” is used to describe the studied compounds. What is so novel about them? They have not been synthesized nor studied for the first time, therefore I can’t see any aspect of their novelty.

The reference [1]. First, it should be updated (replaced) with more recent information on the breast cancer. Second, it is not formatted correctly (Line 538, [1] is missing)

In the introduction, the chemical structures of berberin and berberrubine should be introduced in a form of a Figure.

Line 110, in the future studies, in order to study the mechanism of action of a particular compound, I strongly recommend some in silico molecular modelling methods, i.e. molecular docking combined with molecular dynamics simulations

Line 199, what was the chemical quality of those compounds? Was the purity somehow checked?

Line 243, the authors must decide whether to use “dosage” as here, and LD or concentration and then LC

Figure 2b, how can the authors explain the oscillations of cell viability, exceeding the uncertainties?

Line 289, there is a strange symbol before M, it should be micro or μ

Figure 3, why those changes have not been analyzed using ANOVA?

 Lines 383 and 511, it should be β, not ®.

The number of references (92) is significantly too large, it should be limited

Figure 5C, the y-axis scale should be in the range of 0-1.5 to increase the clarity

In the supplementary materials, Figure S1 is not titled properly, “Figure S1” is missing in its caption. Besides, the Figure S1 and Tables S1-S2 should be in the same file.

Author Response

please find comments to reviewers into the attached file

Reviewer 2 Report

Comments and Suggestions for Authors

The manuscript was ained to examine the anti-timor activity of berberine analogues. 

I have the following suggestions and concerns about this manuscript. 

1) One of the major limitations of the study is about only one cancer cell line  used to examine cytotoxic activities of berberin (BBR) and its analogous. Given that its  was a  canine mammary carcinoma cell lin CF33, all the information about human breast cancer shown in the introduction and discussion sections looks a bit surprising and to my opinion is not relevant to the present study. 

2)  The authors have to include the representative panel of FACs images  (at least for one of the compounds) to illustrate a time dependent increase of early and late apoptotic cells after treatment with BBR analogues. 

3) Another concern about the datashown in Figure 3 is absence of data illustrating the pro-apoptotic effects of the compounds. Indeed, it show just the ratios between early-and late apoptotic cells and did not provide any evidence about the proapoptotic activity of the compounds examined in present study. 

4)  Even ponceau red is shown in western blot images on Figure 4,  expression of the actin, tubulin, etc is usually used for the loading control.  

5) The authors have to describe more precisely the molecular mode of action of the BBR analogues. Given that BBR was shown to induce DNA cleavage (line 100), the authors have to show whether the cytotoxic activities of the BBR analogues is due to its ability to induce DNA damage. To illustrate this possibility, Comet assay is highly recommended. This data should be also supported by WB to illustrate activation of DNA damage pathways in cancer cells and accumulation of H2AX phosphorylated at residue 139 to confirm whether  DNA damage is acquired by cancer cells after treatment with BBR analogues. 

Comments on the Quality of English Language

The manuscript was ained to examine the anti-timor activity of berberine analogues. 

I have the following suggestions and concerns about this manuscript. 

1) One of the major limitations of the study is about only one cancer cell line  used to examine cytotoxic activities of berberin (BBR) and its analogous. Given that its  was a  canine mammary carcinoma cell lin CF33, all the information about human breast cancer shown in the introduction and discussion sections looks a bit surprising and to my opinion is not relevant to the present study. 

2)  The authors have to include the representative panel of FACs images  (at least for one of the compounds) to illustrate a time dependent increase of early and late apoptotic cells after treatment with BBR analogues. 

3) Another concern about the datashown in Figure 3 is absence of data illustrating the pro-apoptotic effects of the compounds. Indeed, it show just the ratios between early-and late apoptotic cells and did not provide any evidence about the proapoptotic activity of the compounds examined in present study. 

4)  Even ponceau red is shown in western blot images on Figure 4,  expression of the actin, tubulin, etc is usually used for the loading control.  

5) The authors have to describe more precisely the molecular mode of action of the BBR analogues. Given that BBR was shown to induce DNA cleavage (line 100), the authors have to show whether the cytotoxic activities of the BBR analogues is due to its ability to induce DNA damage. To illustrate this possibility, Comet assay is highly recommended. This data should be also supported by WB to illustrate activation of DNA damage pathways in cancer cells and accumulation of H2AX phosphorylated at residue 139 to confirm whether  DNA damage is  acquired by cancer cells after treatment with BBR analogues. 

Author Response

please find comments to the reviewers in the attached file

Reviewer 3 Report

Comments and Suggestions for Authors

The article sent to me for review concerns an extremely interesting topic, focused on the use of commonly available natural alkaloids and a set of their semi-synthetic derivatives. These compounds have been subjected to multidirectional tests to demonstrate their anticancer activity in a specific cell model, using for this purpose in vivo studies. This work is prepared in a very careful, exhaustive, factual and current manner.

Berberine used for research is relatively easy to obtain and is therefore a good raw material for obtaining its many semi-synthetic derivatives, which in turn may be an excellent material that can be used in therapy, among others. in the treatment of various cancer diseases. The theoretical assumptions for the work were prepared in a very thoughtful way. And although the selected cell model is not a commonly used one, research planned in such a way is also necessary to combine it with the results of other work to provide a full, comprehensive view of the functional possibilities of the tested substances.

The reviewed article contains all the necessary elements characteristic of a typical original experimental work. They are present in a very balanced quantity, appropriate to the needs. All chapters are prepared in a clear, transparent, readable and easily understandable way. The substantive content of the work also raises no objections. All experiments were correctly planned conceptually, correctly performed in the laboratory and well described. From my point of view, I do not see any elements that require improvement in a scientific sense. The only technical thing that I may have reservations, is the significantly low resolution and therefore low readability of Figure 2.

I recommend that the work be accepted for publication in its current content form, but after obvious editorial correction of a few editorial errors.

Author Response

(The authors gave the same response as above.)

Round 2

Reviewer 1 Report

Comments and Suggestions for Authors

The Authors have corrected and improved their work. This version can be accepted.

Reviewer 2 Report

Comments and Suggestions for Authors

The authors responded to my comments and suggestions. The manuscript is acceptable for publication in present form